# ATR kinase inhibition induces unscheduled origin firing through a Cdc7-dependent association between GINS and And-1

Tatiana Moiseeva [1], Brian Hood[2], Sandy Schamus[1], Mark J. O'Connor[3], Thomas P. Conrads[2] & Christopher J. Bakkenist[1,4]

ATR kinase activity slows replication forks and prevents origin firing in damaged cells. Here we describe proteomic analyses that identified mechanisms through which ATR kinase inhibitors induce unscheduled origin firing in undamaged cells. ATR-Chk1 inhibitor-induced origin firing is mediated by Cdc7 kinase through previously undescribed phosphorylations on GINS that induce an association between GINS and And-1. ATR-Chk1 inhibitor-induced origin firing is blocked by prior exposure to DNA damaging agents showing that the prevention of origin firing does not require ongoing ATR activity. In contrast, ATR-Chk1 inhibitor-induced origins generate additional replication forks that are targeted by subsequent exposure to DNA damaging agents. Thus, the sequence of administration of an ATR kinase inhibitor and a DNA damaging agent impacts the DNA damage induced by the combination. Our experiments identify competing ATR and Cdc7 kinase-dependent mechanisms at replication origins in human cells.

[1] Department of Radiation Oncology, University of Pittsburgh School of Medicine, Hillman Cancer Center, Research Pavilion, Suite 2.6, 5117 Centre Avenue, Pittsburgh, PA 15213-1863, USA. [2] Inova Schar Cancer Institute, Inova Center for Personalized Health, Annandale, VA 22003, USA. [3] DNA Damage Response Biology, Innovative Medicines and Early Clinical Development, Little Chesterford, Saffron Walden, CB10 1XL, UK. [4] Department of Pharmacology and Chemical Biology, University of Pittsburgh, Pittsburgh, PA 15261, USA. Correspondence and requests for materials should be addressed to C.J.B. (email: bakkenistcj@upmc.edu)

Pharmacologic DNA damage response (DDR) inhibitors that selectively inactivate the enzymatic activities of ATM, ATR, Chk1, Chk2, DNA-PK, PARP, PARG, and Wee1 have the potential to increase the efficacy of standard-of-care chemotherapy[1]. The sequence of administration of a DDR inhibitor and an agent that directly damages DNA may impact the efficacy of cell killing by the combination. If the DDR inhibitor is administered after the DNA damaging agent, the consequences of inhibiting the activity of the DDR enzyme will be focused on tolerance and repair mechanisms. In the majority of preclinical experiments, however, the DDR inhibitor is administered before the DNA damaging agent and the physiological consequences of inhibiting the activity of the DDR enzyme in unperturbed cells are not considered. This is inappropriate as DDR inhibitors may cause changes in DNA metabolism and/or cell cycle progression in unperturbed cells that impact the number of lesions induced by the subsequent administration of a DNA damaging agent.

ATR (ataxia telangiectasia and Rad3-related) is an essential DDR kinase in mice and mammalian cell lines and this has been attributed to activities associated with DNA replication[2, 3]. ATR kinase is activated at stalled and collapsed replication forks and phosphorylates several thousand protein substrates that function in DNA replication and repair, chromatin remodeling, transcription, protein synthesis and degradation, cell cycle progression, and cell death[4, 5]. ATR phosphorylates and activates Chk1, a second essential DDR kinase that phosphorylates protein substrates at a different consensus motif[6, 7]. The complexity of identified DNA damage signaling indicates that >10% of the proteome may be modified by ATR kinase-dependent signaling after exposure to clinically relevant doses of DNA damaging chemotherapy, ionizing radiation (IR), or ultraviolet radiation (UV). Modifications embedded within this complex system induce the DNA replication checkpoint that protects stalled and collapsed replication forks and inhibits DNA replication origin firing. This pan-nuclear inhibition of origin firing is caused, at least in part, by ATR and Chk1 kinase-dependent phosphorylation and degradation of Cdc25A, a phosphatase that is required to dephosphorylate and activate Cdk2, a kinase essential for origin firing[8, 9].

ATR kinase activity is also implicated in the regulation of unperturbed DNA replication by recent observations that ATR kinase inhibition induces unscheduled origin firing and reduces replication fork velocity by an unknown mechanism(s)[10, 11]. To our knowledge, these observations have not been pursued further. The human genome is replicated by ~50,000 replicons of 100,000 bp[12]. Replication is initiated in these replicons in a temporally ordered sequence through S phase and ~10% of replicons are engaged in DNA synthesis at any given time. The pre-replication complex (pre-RC) is generated in G1 phase by the sequential loading of the origin recognition complex, licensing factors Cdt1 and Cdc6, and the inactive minichromosome maintenance (MCM) core helicase complex onto chromatin[13]. The number of pre-RCs loaded during G1 phase, from yeast to humans, greatly exceeds the number of replication origins that fire in unperturbed S phase[14]. Little is known about the mechanisms that regulate the temporal activation and suppression of potential replication origins, but dormant origins can be activated following replication stress and this may ensure the completion of DNA replication[15–17]. Whether the firing of dormant origins is a regulated event, or a stochastic event afforded by the increased opportunity for these dormant origins to fire is unclear.

Origin firing requires the assembly of Cdc45, MCM, and GINS proteins and activation of the replicative helicase (CMG) at pre-RC[18]. In yeast, this assembly requires a series of additional factors including Sld2, Sld3, Sld7, Dpb11, and MCM10 that act via formation of a transient pre-initiation complex (pre-IC). Cdc7 and Cdk2 kinase activities are required to activate CMG helicase and initiate DNA unwinding[18, 19]. In yeast, a complex of DNA polymerase alpha and Ctf4/And-1(WHDH1/And-1 in humans) interacts with GINS in the CMG[20], but it has never been demonstrated in mammalian cells.

We have investigated the mechanism through which ATR kinase inhibitors (ATRi's) induce unscheduled origin firing. We show that ATR and Chk1 kinase inhibition induce a dramatic accumulation of replication-associated proteins and hyper-phosphorylation of the replicative helicase subunit MCM4 in the nuclease-insoluble chromatin proteome and the association of And-1 with GINS in a Cdc7 kinase-dependent manner. We also demonstrate here that Chk1 kinase activity and Chk1 kinase activation are ATR kinase-dependent and that ATR kinase inhibitors block both ATR and Chk1 kinase activities in cells with similar kinetics. The effects of ATRi's and Chk1 kinase inhibitor (Chk1i) are blocked when DNA damage is induced prior to ATR kinase inhibition. Our results could have significance as pharmacologic ATR kinase inhibitors AZD6738[21] and VX-970[22] are in clinical trials.

## Results

**ATRi induces unscheduled origin firing**. ATR kinase activity is implicated in the regulation of unperturbed DNA replication by recent observations that ATR kinase inhibition induces unscheduled origin firing and reduces replication fork velocity[10, 11]. We used DNA combing system to study DNA replication in 293T cells treated with vehicle or ATR inhibitor. As expected, ATRi AZD6738 reduced inter-origin distance and the length of labeled DNA tracks (Fig. 1a–c). ATR kinase inhibition did not induce replication fork asymmetry in unperturbed cells (Supplementary Fig. 1).

To investigate the mechanism of ATRi-induced unscheduled origin firing, we isolated and identified the nuclease-insoluble chromatin proteome in 293T cells treated with vehicle or ATRi AZD6738 using high-resolution liquid chromatography-tandem mass spectrometry (LC-MS/MS) (Supplementary Data 1). Previous unbiased surveys of ATR kinase signaling have been limited to the soluble proteome and/or have not documented the impact of ATRi on unperturbed cells[5]. ATRi induced an accumulation of polE1, MCM2, MCM7, PSF3, Timeless, DNA polymerase alpha, and And-1 in the chromatin proteome (Fig. 1d). While ATR and RPA34 accumulate in the chromatin after treatments with DNA damaging agents, neither of these proteins were recruited to the chromatin in cells treated with ATRi alone (Fig. 1e). Neither DNA polymerase delta nor PCNA were recruited to the chromatin and the abundance of histone proteins in the chromatin proteome did not change in unperturbed cells treated with ATRi.

To confirm that ATRi induced an accumulation of replication proteins at active replication forks, we purified EdU-labeled nascent DNA using the iPOND technique[23, 24]. The association of PSF3, a component of GINS complex, with nascent DNA was increased in cells treated with ATRi (Fig. 1f). Histone H3 served as a control for the purification of nascent DNA. The replisome components that accumulated in the chromatin after ATRi treatment are summarized in a schematic (Fig. 1g). These data show that ATR kinase activity is essential to prevent the recruitment of replication proteins to chromatin.

**ATRi and Chk1i induce hyper-phosphorylation of MCM4**. We observed a modified MCM4 protein with decreased mobility in SDS-PAGE in the chromatin proteome of unperturbed cells treated with ATRi AZD6738 (Fig. 1d). This modification of MCM4 was induced in cells treated with the selective, clinical

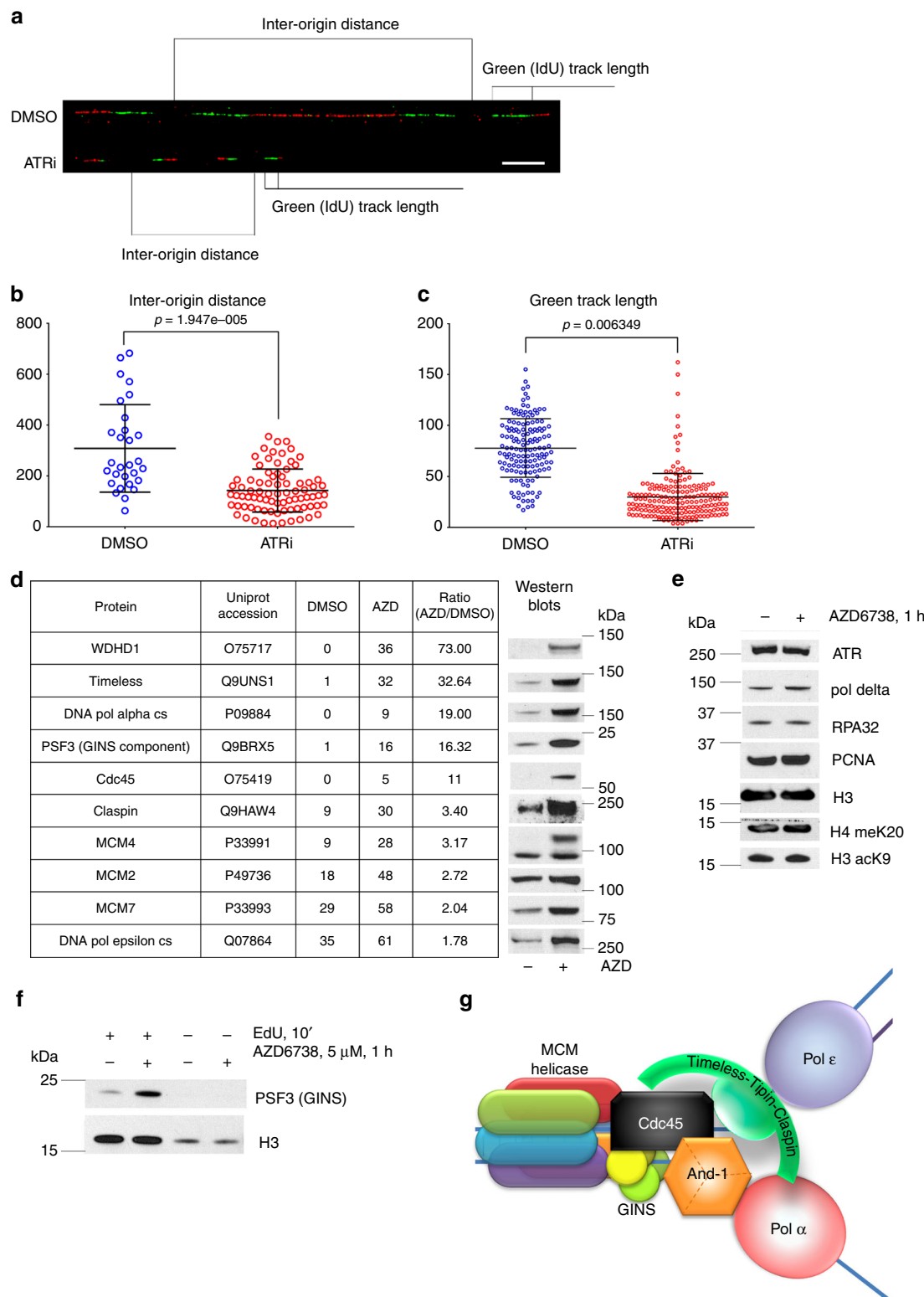

**Fig. 1** ATR inhibition induces origin firing and an accumulation of the replication proteins in the insoluble chromatin fraction. **a** 293T cells treated with DMSO/5µM AZD6738 for 30' before addition of IdU for 10' followed by incubation with CldU for 20'. Newly replicated DNA was analyzed by molecular combing, representative images are shown (IdU in green, CldU in red). Scale bar represents 10 µm. **b** The inter-origin distances on molecular combing images were measured and analyzed. **c** The lengths of IdU (green) tracks on molecular combing images were measured and analyzed. For **b** and **c**, error bars represent standard deviation. **d**, **e** 293T cells were treated with AZD6738 or DMSO for 30 min and then fractionated. Nuclease-insoluble fraction was analyzed by mass spectrometry showing increase of the proteins involved in replication. Western blots confirm the MS results. **f** After 1 h treatment with 5 µM AZD6738, newly synthesized DNA was labeled with EdU for 10 min in presence of AZD6738 and pulled down with click chemistry protocol after cross-linking the proteins to DNA. Western blot analysis of the replication associated PSF3 protein (GINS component) is shown. **g** Schematic picture of the basic replication fork components

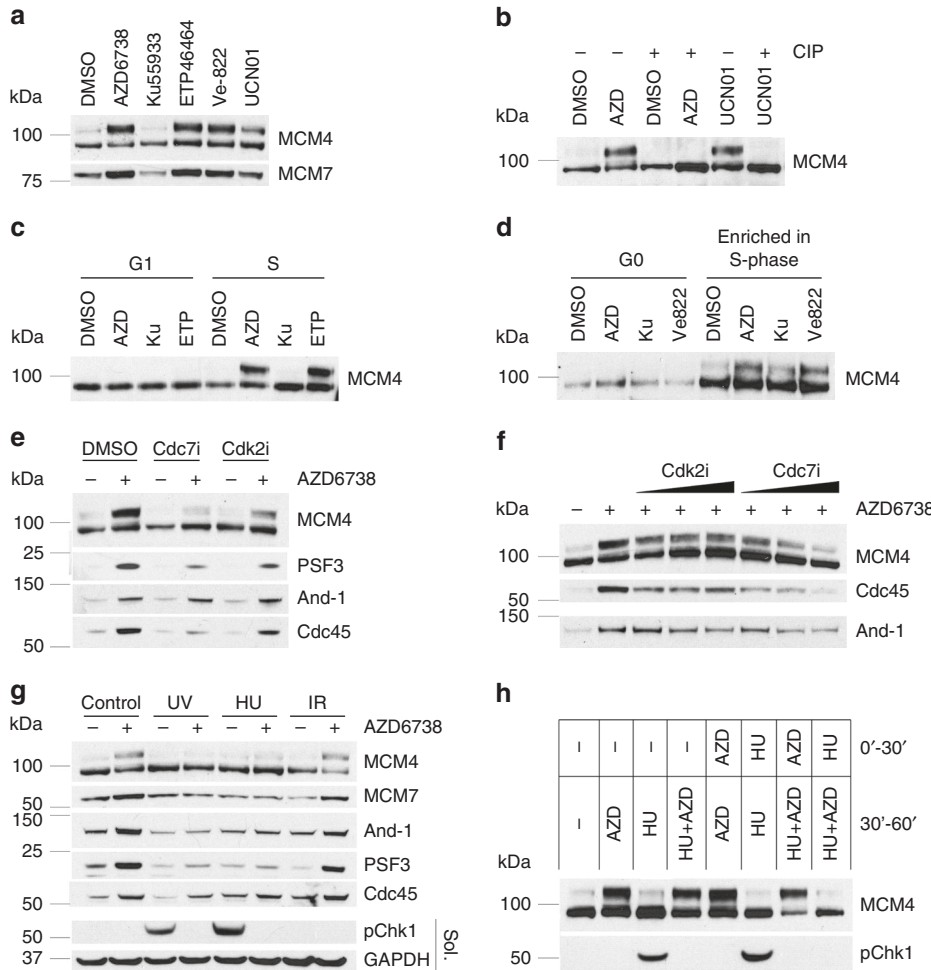

**Fig. 2** ATR inhibition causes Cdc7-dependent MCM4 hyperphosphorylation in the chromatin fraction of S-phase cells. Cells were fractionated as described in Methods section. Western blotting of the nuclease-insoluble chromatin fraction is shown. **a** 293T cells were treated with 5 μM AZD6738, 10 μM Ku55933, or 5 μM ETP46464, 5 μM Ve-822, or 100 nM UCN01 for 1 h. **b** 293T cells were treated with 5 μM AZD6738 or 100 nM UCN01 for 1 h, the nuclease-insoluble fraction was treated with CIP or just incubated at 37 °C. **c** U2OS cells were synchronized by double thymidine-nocodazole block and then released into mitosis. G1 phase cells (6 h later) or S-phase cells (12 h later) were treated with 5 μM AZD6738, 5 μM ETP46464, 10 μM Ku55933, or DMSO for 1 h. **d** BJ-hTERT cells were synchronized by contact inhibition and then released by splitting (for S phase 10 h later) or kept confluent (for G1 phase). Cells were treated with 5 μM AZD6738, 5 μM Ve-822, 10 μM Ku55933, or DMSO for 1 h. **e** Cdc2 or Cdc7 inhibitor of 10 μM were added to 293T cells 15 min before 1 h treatment with 5 μM AZD6738. After fractionation, the nuclease-insoluble fraction was analyzed by western blotting. **f** Increasing concentrations (2, 5, 10 μM) of Cdk2 or Cdc7 inhibitor were added to 293T cells 15 min before 1 h treatment with 5 μM AZD6738. After the fractionation, the nuclease-insoluble fraction was analyzed by western blotting. **g** 293T cells were treated with 5 Gy IR, 10 J m$^{-1}$ UV or 5 mM HU 30 min before the addition of 5 μM AZD6738 for another 30 min. **h** Picture indicates the conditions for the first 30 min of treatment and for the second 30′ of treatment. After the second 30 min, treatment cells were fractionated and analyzed

pharmacologic ATRi's AZD6738 and Ve822, the ATRi ETP46464[25], and the Chk1 kinase inhibitor UCN01, but not the ATM kinase inhibitor KU55933 (Fig. 2a). The ATRi-induced MCM4 mobility shift was dependent on the concentration of AZD6738 and UCN01 (Supplementary Fig. 2a). Incubation of the chromatin proteome with calf intestinal phosphatase reversed the ATRi-induced MCM4 mobility shift revealing it to be a consequence of phosphorylation (Fig. 2b). Hyper-phosphorylation of MCM4 can be detected as early as 15 min after the start of the ATRi or Chk1i treatment (Supplementary Fig. 2b). We also showed that ATR inhibition causes rapid Chk1 inactivation (Supplementary Fig. 2c), which explains why this response to ATRi and Chk1i is similar.

To determine whether the ATRi-induced hyper-phosphorylation of MCM4 in the chromatin proteome is restricted to replicating cells, we synchronized U2OS cells using a double thymidine-nocodazole block and BJ-hTERT fibroblasts using contact inhibition. ATRi-induced hyper-phosphorylation of MCM4 was only observed in S-phase cells (Fig. 2c, d). These data show that ATR kinase activity prevents hyper-phosphorylation of MCM in the chromatin proteome of unperturbed normal and cancer cells.

**ATRi-induced hyper-phosphorylation of MCM4 is Cdc7-dependent.** To determine whether the ATRi-induced hyper-phosphorylation of MCM4 and the accumulation of replication proteins in the chromatin proteome of unperturbed cells are Cdc7 and/or Cdk2 kinase-dependent, we pre-treated cells with Cdc7 kinase inhibitor PHA-767491 or Cdk2 inhibitor CVT313 before the addition of ATRi AZD6738. It has been shown previously that Chk1i-induced unscheduled origin firing is both Cdc7 and Cdk2 kinase-dependent[26].

Cdc7 kinase inhibitor totally blocked the ATRi-induced hyper-phosphorylation of MCM4 and the accumulation of PSF3 and

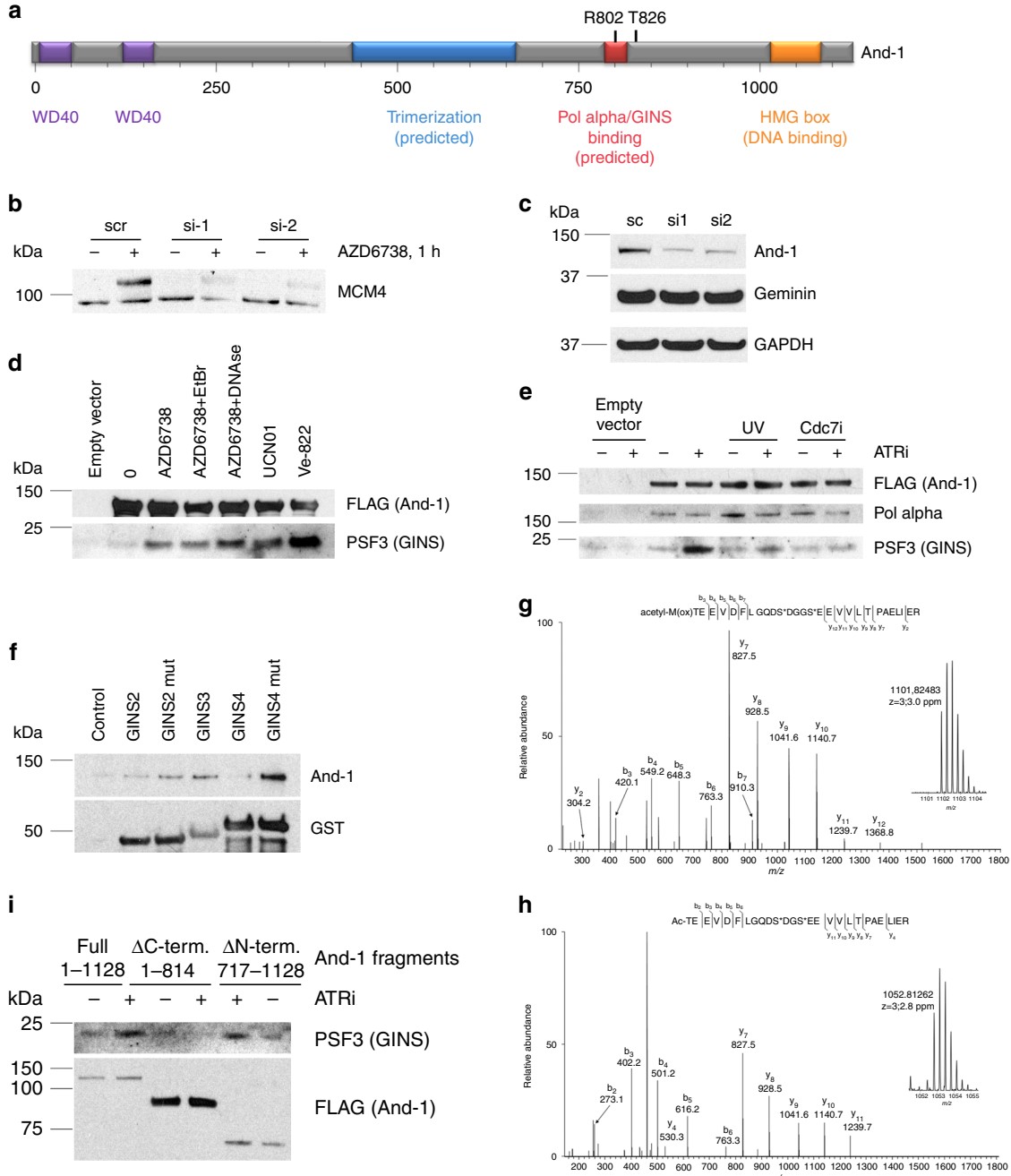

**Fig. 3** ATR inhibition causes an increased Cdc7-dependent association between And-1 and GINS. **a** Schematic representation of And-1 structure and functions. **b** 293T cells were transfected with siRNAs against And-1. Forty-eight hours later, cells were treated with vehicle or AZD6738 for 1 h and fractionated. Western blot of the nuclease-insoluble fraction is shown. **c** Whole-cell lysate western blot of 293T cells transfected with siRNAs against And-1. **d** 293T cells were transfected with empty vector or vector expressing And-1-FLAG. Forty-eight hours later, cells were treated with 5 μM ATRi (AZD6738), 200 nM Chk1i (UCN01), or 5 μM ATRi (Ve822). Cells were lysed, and And-1-FLAG was immunoprecipitated and eluted with FLAG peptide. EtBr or DNAse I were added to the indicated samples during the IP. **e** 293T cells were transfected with empty vector or vector expressing And-1-FLAG. Forty-eight hours later, cells were treated with 5 μM ATRi (AZD6738). Indicated samples were UV-irradiated at 10 J m⁻² 30′ before ATRi, Cdc7i was added 15′ before ATRi. Cells were lysed, And-1-FLAG was immunoprecipitated and eluted with FLAG peptide. **f** Wild-type or mutant GINS proteins tagged with GST were expressed in *E.coli*, purified using glutathione sepharose. GINS proteins attached to beads were incubated with total cell lysate of 293T cells, washed with lysis buffer, and eluted with glutathione solution. Analysis of the eluted samples is shown. **g, h** Liquid chromatography-tandem mass spectrometry (LC-MS/MS)-based identification of two N-terminal phosphorylated structural isoforms of GINS4. **g** Tandem MS and accurate mass (inset *m/z* 1101.82483, charge state 3+) of the phosphorylated form of the methionine (oxidized)-containing N-terminal tryptic peptide of GINS4. **h** Tandem MS and accurate mass (inset *m/z* 1052.81262, charge state 3+) of the phosphorylated form of the methionine cleaved N-terminal tryptic peptide of GINS4. **i** 293T cells were transfected with empty vector or vector expressing full size And-1-FLAG or specified truncation mutants. Forty-eight hours later, cells were treated with 5 μM ATRi (AZD6738) for 1 h. Cells were lysed, And-1-FLAG was immunoprecipitated and eluted with FLAG peptide. Eluted samples were analyzed by western blotting

Cdc45 in the chromatin (Fig. 2e). Cdk2 kinase inhibitor only partially inhibited the ATRi-induced accumulation of PSF3 and Cdc45 in the chromatin proteome of unperturbed cells. Furthermore, Cdc7i blocked ATRi-induced hyper-phosphorylation of MCM4 and the accumulation of PSF3 and Cdc45 in the chromatin in a dose-dependent fashion (Fig. 2f). These data show that ATR kinase activity is essential to prevent Cdc7 kinase-dependent hyper-phosphorylation of MCM4 and the accumulation of replication proteins in the chromatin of unperturbed cells. Cdk2i and Cdc7i at the concentrations completely blocked DNA replication at 12 h after addition as determined by EdU incorporation (Supplementary Fig. 2d). The broad spectrum CDK inhibitor roscovitine also blocked ATRi-induced MCM4 hyperphosphorylation (Supplementary Fig. 2e), but the identity of the relevant target of roscovitine was not explored further.

**DNA damage blocks ATRi-induced unscheduled origin firing**. MCM4 is phosphorylated in yeast by Cdc7 (DDK) and in yeast and mammalian cells by Cdk2 and these phosphorylations are associated with activation of CMG helicase[18, 19, 27–29]. Hyper-phosphorylation of MCM4 in the chromatin may be an excellent marker for unscheduled origin firing in mammalian cells.

To determine the impact of DNA damage on ATRi-induced hyper-phosphorylation of MCM4, we treated cells with IR, UV, or hydroxyurea (HU) 30 min before the addition of ATRi for an additional 30 min. Treatment with UV and HU, but not IR, suppressed ATRi-induced hyper-phosphorylation of MCM4 in the chromatin (Fig. 2g). In contrast, treatment with HU 30 min after ATRi did not suppress ATRi-induced hyper-phosphorylation of MCM4 (Fig. 2h). These data show that ATR and Chk1 kinase activities are not required to suppress hyper-phosphorylation of MCM4 in the chromatin proteome of cells after treatment with DNA damaging agents inducing replication checkpoint.

**ATRi and Chk1i induce an association between GINS and And-1**. The accumulation of WDHD1/And-1 in the chromatin proteome of unperturbed cells treated with ATRi was the greatest ATRi-induced protein dynamic identified by MS (Fig. 1d). Ctf4/And-1, the yeast homolog of human And-1, is a homotrimer that interacts with DNA polymerase alpha and Sld5 (GINS component)[20]. Ctf4 is believed to recruit DNA polymerase alpha to CMG[20]. Human And-1 is 250 amino acids longer than yeast Ctf4/And1 and this additional sequence includes a HMG box important for DNA binding (Fig. 3a).

To investigate the role of And-1 in ATRi-induced unscheduled origin firing, we used two different small interfering RNAs (siRNAs) to knock down And-1 in 293T cells. And-1 disruption suppressed the ATRi-induced hyper-phosphorylation of MCM4 in the chromatin of unperturbed 293T cells (Fig. 3b, c). This was not a consequence of decreased replication as the levels of geminin and the percentage of cells in S phase did not change. The interaction between And-1 and PSF3 (GINS) was increased in cells treated with ATRi AZD6738, ATRi Ve822, and Chk1i UCN01 (Fig. 3d). This interaction was not mediated by DNA as it was insensitive to treatment with either ethidium bromide or DNase I. The interaction between And-1 and PSF3 (GINS) induced by ATR kinase inhibition was blocked by prior exposure to UV or Cdc7 kinase inhibitor (Fig. 3e). The interaction between And-1 and DNA polymerase alpha, however, was not changed by ATR kinase inhibition, UV, or Cdc7 kinase inhibition (Fig. 3e). These data suggest that ATR kinase and Chk1 kinase inhibition cause unscheduled dormant origin firing, at least in part, by promoting a Cdc7 kinase-dependent interaction between And-1 and GINS complex. According to our model,

ATR and Chk1 kinase activity prevent the recruitment of And-1-DNA polymerase alpha to GINS in pre-RCs at dormant origins in the vicinity of the ongoing replication.

**Cdc7 phosphorylates GINS4 to induce association with And-1**. The interaction of Ctf4 (And-1) with Sld5 (GINS4) in yeast was previously described[20]. However, there's very little sequence similarity between yeast and human Sld5 proteins. The peptide, crystalized in complex with yeast Ctf4 (shown in red frame), is absent in human Sld5 (Supplementary Fig. 3a). We propose that the interaction between GINS and And-1 in human cells is more complex and requires Cdc7 phosphorylation, while in yeast this step of origin firing is not regulated. The potential Cdc7 phosphorylation sites on Sld5 are shown with arrows on Supplementary Fig. 3a. It is also possible that in human cells a subunit other than Sld5 binds And-1.

In order to further investigate this, we purified GST-fused GINS subunits from bacteria using glutathione sepharose beads and used them as baits to examine which protein is capable of pulling down endogenous And-1 from 293T cell lysate (Fig. 3f, g). Since Cdc7-dependent phosphorylation may be required for GINS-And-1 interaction, known phosphorylation sites suspected to be Cdc7-dependent (S/T E, S/T D)—S104 on GINS2, and S12 and S16 on GINS4—were mutated to glutamate to mimic phosphorylation, and their ability to pull down And-1 from cell lysates was compared to that of the wild-type proteins. This experiment showed that GINS2, 3, and 4 proteins interacted with And-1, and phosphomimetic mutations on both GINS2 and GINS4 positively affected these interactions (Fig. 3f). Interpretation of these data is complicated because the interaction of GST-fused GINS protein with And-1 might be mediated by other GINS subunits present in human cell lysate.

To identify GINS4 phosphorylation(s), GINS4-FLAG was immunoprecipitated from 293T cells treated with vehicle, ATRi AZD6738, and the combination of ATRi AZD6738 and Cdc7i, digested with trypsin, and analyzed by LC-MS/MS. MS identified structural isoforms of the N-terminal peptide with and without the N-terminal methionine (Supplementary Fig. 3b–g). Both of these N-terminal peptide isoforms were phosphorylated (Fig. 3g, h). While the fragmentation generated by collision-induced dissociation of these peptides did not enable the site of phosphorylation to be localized to S12 or S16, the data unambiguously show that GINS4 is phosphorylated in cells treated with ATRi AZD6738 on either S12 or S16. Taken together, the mass chromatograms generated from the N-terminal GINS4 peptide (met-present, met-cleaved, and each phosphorylated) suggests that the phosphorylated form is increased in cells treated with ATRi and that this increase is blocked by Cdc7i (Supplementary Fig. 3b–g), however, these changes proved challenging to accurately quantify in subsequent analyses.

**The C-terminus of And-1 interacts with GINS**. Human And-1 and yeast Ctf4 have more sequence similarity than GINS proteins, however, And-1 has an additional ~250 amino acid C-terminal region that is absent in yeast. We hypothesize that this part may participate in the regulation of And-1-GINS interaction. In order to test that, we expressed truncated versions of And-1 corresponding to amino acids 1-814 (mimicking yeast Ctf4) and 717-1128 (C-terminal part) along with the wild-type And-1, and tested the co-immunoprecipitation of GINS subunit PSF3 with them in response to ATR inhibitor treatment for 1 h (Fig. 3i). C-terminal deletion completely abrogated GINS recruitment to And-1 after ATR inhibition, showing that C-terminal fragment, absent in yeast, is critical for And-1 interaction with GINS. In agreement with these data, FLAG-tagged C-terminal fragment of

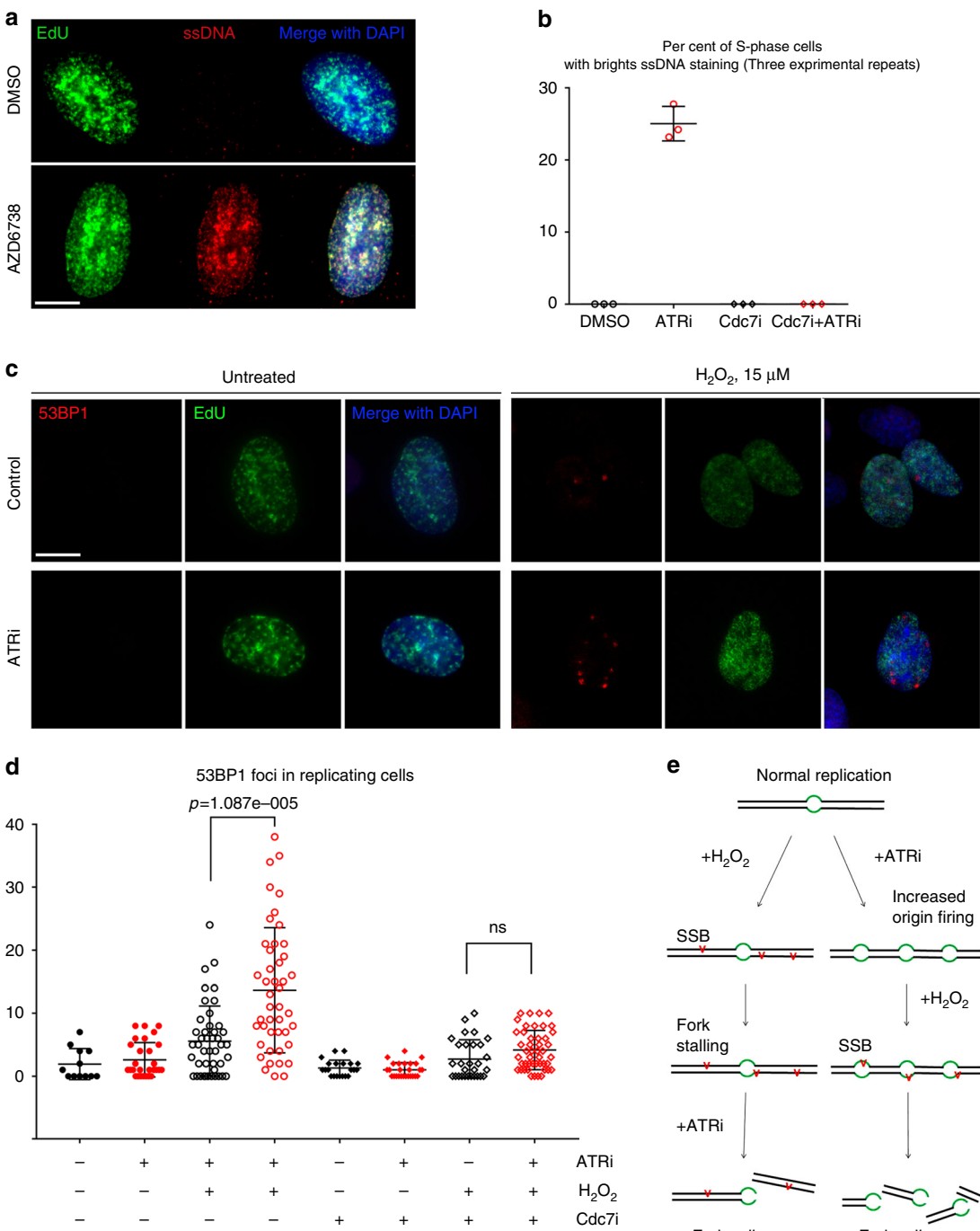

**Fig. 4** Prior ATR inhibition increases the number of DNA double-strand breaks induced by hydrogen peroxide. **a** U2OS cells were incubated with IdU for 24 h before the experiment. EdU of 10 μM was added 15′ before 1 h treatment with 5 μM AZD/DMSO. Cells were fixed and stained with anti-BrdU antibody to detect ssDNA followed by click-chemistry reaction to visualize EdU. Scale bar represents 10 μm. **b** Quantification of the previous experiment. For Cdc7i samples, 10 μM Cdc7i was added at the same time as EdU. Error bars represent standard deviation. **c** BJ-hTERT fibroblasts were treated with vehicle or 5 μM ATRi (AZD6738) for 45 min before the addition of 15 μM $H_2O_2$ and 10 μM EdU to label replicating cells. Cells were fixed 30 min after adding $H_2O_2$. 53BP1 was detected by indirect immunofluorescence; EdU was detected by click-chemistry reaction. Representative images are shown. Scale bar represents 10 μm. **d** As in **c**, but Cdc7 inhibitor was added 15 min before ATRi. Numbers of 53BP1 foci per cell were determined. Three experimental repeats were performed and showed similar results; quantification of one is shown. **e** Schematic representation of a model: sequence of administration of ATRi and DNA damaging agent affects the number of induced lesions

And-1 alone was able to pull down PSF3 after AZD6738 treatment. The C-terminal deletion did not affect the ability of And-1 to bind pol alpha, suggesting that it folded correctly (Supplementary Fig. 3h). These data show that And-1 is recruited to GINS component of the CMG helicase in human cells through its C-terminal domain.

**ATRi causes ssDNA accumulation in replicating cells**. Replication fork stalling is frequently associated with an accumulation of single-stranded DNA (ssDNA) and this induces an ATR kinase-dependent DDR[30]. To determine whether ATRi induced ssDNA via a Cdc7 kinase-dependent mechanism, we used a method described previously[31]. We treated cells with IdU for 24 h

to label genomic DNA throughout the asynchronous population and identified IdU in ssDNA regions of chromosomes using an anti-IdU antibody that recognize IdU in ssDNA, but not double-stranded DNA, and conditions that did not denature the DNA. Treatment with ATRi for 1 h caused an accumulation of ssDNA in 25% of replicating cells and this ssDNA co-localized with nascent DNA that was identified by the EdU pulse (Fig. 4a). The ATRi-induced accumulation of ssDNA was blocked by Cdc7 kinase inhibitor (Fig. 4b). And-1 knockdown by siRNA also decreased the number of replicating cells with bright ssDNA staining (Supplementary Fig. 4a, b). We also confirmed that And-1 siRNA knockdown in U2OS was more efficient than in 293T cells, and modestly decreased the number of S-phase cells (Supplementary Fig. 4a, c–f) while still allowing quantification of ssDNA. These data show that ATR kinase activity is essential to prevent an And-1- and Cdc7 kinase-dependent accumulation of ssDNA in replicating cells.

**ATRi increases the number of lesions induced by H$_2$O$_2$.** Our data show that ATR inhibition induces massive unscheduled origin firing and accumulation of ssDNA in replicating cells. Therefore, we hypothesized that the increased number of replication forks and increased amount of ssDNA would cause single-strand breaks to turn into double-strand breaks (DSBs). We chose hydrogen peroxide as an agent normally producing mostly single-strand breaks at a low concentration, and observed 53PB1 foci as a readout for DSBs. BJ-hTERT fibroblasts were treated with AZD6738 or vehicle for 45 min and then 15 μM H$_2$O$_2$ was added to induce DNA damage. (Fig. 4c, d). 53BP1 foci were scored exclusively in the replicating cells as determined by EdU incorporation. This experiment showed a significant increase in the number of 53BP1 foci per replicating cell induced by hydrogen peroxide in the cells, pre-treated with ATR inhibitor (Fig. 4c, d). In order to confirm that this increase in the number of DSB is due to unscheduled origin firing, we added Cdc7 inhibitor to prevent ATRi-induced replication initiation, and this rescued the ATRi-pretreatment-induced increase of DSB after hydrogen peroxide treatment (Fig. 4d).

## Discussion
This manuscript documents, to our knowledge, the first unbiased screen of the impact of ATR kinase inhibition on the chromatin proteome in unperturbed cells and the discovery of a mechanism that suppresses unscheduled origin firing in human cells. Studying the effect of ATR inhibition allowed us to detect a Cdc7-dependent interaction between And-1 and GINS complex in human cells and other markers of origin firing such as MCM4 hyper-phosphorylation and the accumulation of replication fork component in the chromatin.

We propose that the hyper-phosphorylation of human MCM4 in the chromatin is a marker that can be used to study origin firing. It is likely that CMG helicase activation and assembly of the active replisome is associated with topological changes that render the local DNA resistant to nuclease treatment. The hyper-phosphorylation of human MCM4 is restricted to S-phase cells, is Cdc7 kinase-dependent, and is associated with both the recruitment of replication-associated proteins to the chromatin and unscheduled origin firing. While this manuscript was in preparation, hyper-phosphorylation of human MCM4 was also observed by others[32]. Cdc7 and Cdk2 kinase activities are required to activate CMG helicase and initiate DNA unwinding[18, 19]. In yeast, Dbf4-dependent Cdc7 kinase (DDK) phosphorylates several subunits of the MCM2-7 hexamer and DDK-dependent phosphorylations on the amino terminus of MCM4 reverse an intrinsic inhibitory activity[27, 33, 34]. Our data therefore extend

those reported in yeast where hyper-phosphorylation of MCM4 has been associated with the initiation of DNA replication and used as a marker to study mechanisms that impact origin firing[28].

The replication checkpoint induces pan-nuclear inhibition of origin firing, at least in part, by ATR and Chk1 kinase-dependent phosphorylation and degradation of Cdc25A, a phosphatase that is required to dephosphorylate and activate Cdk2, a kinase essential for origin firing[8, 9]. Our data show that Cdk2 kinase inhibition only partially blocks unscheduled origin firing in unperturbed cells treated with ATRi, while Cdc7 kinase inhibitor completely blocks MCM4 hyper-phosphorylation in cells treated with ATRi. Cdk2 activity is required for Cdc7-dependent CMG helicase activation in yeast[35]. However, Cdc7 activity is required for Cdk2-dependent CMG helicase activation in Xenopus[36]. To our knowledge, no conclusive data on this sequence are available in human systems.

Unscheduled origin firing in cells treated with ATRi's and Chk1i is caused by Cdc7 kinase activity at pre-RC. Cdc7 purifies as a heterodimer with Dbf4, an ATR and Chk1 substrate[37, 38]. However, there's no evidence that ATR or Chk1-dependent phosphorylation can regulate DDK activity or stability. There are several possible explanations for increased Cdc7 activity in cells treated with ATR or Chk1 inhibitors. First, DDK may move from stalled forks induced by replication stress to pre-RC at dormant origins. Second, the increased activity may be associated with the second Cdc7 heterodimer, Cdc7-Dbf4b/Drf1. Drf1 is not essential for cell survival (unlike Dbf4), but its knockdown leads to a prolonged S phase that may indicate a role in the DDR[39]. However, our data (Supplementary Fig. 5) suggests that the canonical Dbf4 is responsible for Cdc7-dependent phosphorylations at the replication origins. Third, the increased phosphorylations may be explained by inactivation or mislocalization of a protein phosphatase that reverses Cdc7 kinase-dependent phosphorylations. For example, Rif1 localizes PP1 late origins[32, 40] and Rif1 was identified as a Chk1 substrate[41]. Chk1 kinase-dependent phosphorylation of Rif1 at dormant origins may recruit PP1 and prevent dormant origin firing, and this mechanism may be reversed by ATR or Chk1 inhibition. Mechanisms that localize Cdc7 kinase activity are being investigated further.

We show that ATR and Chk1 kinase activities are essential to prevent Cdc7 kinase-dependent hyper-phosphorylation of MCM in the chromatin fraction of unperturbed cells. Strikingly, after treatment with DNA damaging agents that target DNA replication, ATR and Chk1 kinase activities are no longer required to prevent hyper-phosphorylation of MCM in the chromatin fraction. The ATR kinase signaling induced by DNA damaging agents is totally blocked by ATRi's administered after the initiation of DNA damage, but in these damaged cells, ATRi's do not induce hyper-phosphorylation of MCM4. These data imply the existence of an early mechanism induced by DNA damaging agents that cannot be reversed by ATR inhibition.

Since preliminary DNA damage had the same effects on unscheduled origin firing as Cdc7 kinase inhibitor, we propose that an ATR and Chk1 kinase-dependent mechanism induced in cells prevents Cdc7 from phosphorylating its substrates at the pre-RC. In yeast, replication stress blocks origin firing through Rad53-dependent phosphorylation of Dbf4[42] and Sld3[43]. However no similar mechanisms have been described in human systems. Cdc7-Dbf4 (DDK) complex is neither degraded nor inhibited after DNA damage or during DNA replication in human cells[37, 44, 45]. DNA damage-induced ATR and Chk1 kinase activities may mislocalize Cdc7 kinase thereby blocking its activity at pre-RC. Alternatively, DNA damage-induced ATR and Chk1 kinase activities may degrade a protein that recruits Cdc7 to pre-RC. It is well established that Cdt1, an essential component of pre-RC, is rapidly degraded after DNA damage[46]. Cdt1 interacts

with Cdc7 and is important for Cdc45 and MCM recruitment to pre-RC[47, 48]. Therefore, Cdt1 degradation after DNA damage is a potential mechanism through which Cdc7 kinase activity at pre-RC could be blocked. We have confirmed that Cdt1 is degraded 1 h after UV treatment, as expected (Supplementary Fig. 5). However, Cdt1 was not degraded 1 h after either HU or IR. These data show that an ATR kinase-dependent, Cdt1-independent, mechanism induced in cells after DNA damage suppresses Cdc7 activity at pre-RCs.

And-1 interacts with the p180 subunit of DNA polymerase alpha and is essential for its stability in human cells[49]. We show that the interaction between And-1 and GINS is increased in unperturbed cells treated with ATRi's and Chk1i. This interaction increase is completely blocked by Cdc7 kinase inhibition or DNA damage that targets DNA replication. The Cdc7 kinase-dependent interaction between And-1 and GINS is a previously unidentified step in the activation of pre-RC and origin firing. Our data also identifies C-terminal part of And-1, absent in yeast Ctf4, as responsible for interaction with GINS. This implies that And-1 and GINS step of origin firing is significantly different in human cells compared to yeast, and calls for further investigation of human origin firing mechanisms.

The sequence of administration of an ATR kinase inhibitor and a DNA damaging agent impacts the amount of cytotoxic DNA damage induced by the combination. We show that when an ATRi is administered after the DNA damaging agent, the consequences of inhibiting the activity of ATR will be focused on tolerance and repair mechanisms. However, if the ATRi is administered before the DNA damaging agent then unscheduled origin firing will be induced prior to DNA damage and there will be more DNA replication forks when the DNA damaging agent is administered. Since cancer cells acquire mutations that change DNA metabolism and/or cell cycle progression, the administration of a DDR inhibitor prior to the DNA damaging agent may have a different impact in normal and cancer cells leading to increased therapeutic index and efficacy of the combination. Cancer cells with replicative stress as a result of oncogene activation or loss of tumor suppressor checkpoint mechanisms are likely to be more sensitive to unscheduled origin firing as the capacity of the replication machinery is compromised. Our finding that the sequence of administration of an ATR kinase inhibitor and a DNA damaging agent impacts the DNA damage induced by the combination.

## Methods

**Cell lines and synchronization.** HEK293T cells and BJ-hTERT fibroblasts were cultured in DMEM and U2OS in RPMI media containing 10% FBS, 100 U ml$^{-1}$ penicillin and 100 mg ml$^{-1}$ streptomycin (Lonza). All the cell lines were purchased from ATCC. For synchronization, U2OS cells were blocked in S-phase with 2 mM thymidine for 24 h, released for 4 h, and blocked in M phase with 100 ng ml$^{-1}$ nocodazole.

**Antibodies and chemicals.** Antibodies: WDHD1(Abcam #67341) (used at 1:500), timeless (Abcam #50943) (used at 1:400), DNA pol alpha cs (Abcam #176734) (used at 1:2000), PSF3 (Abcam #177515), Cdc45 (Cell Signaling #3673) (used at 1:1000), Claspin (Cell Signaling #2800) (used at 1:1000), MCM4 (Cell Signaling #3228) (used at 1:1000), MCM2 (Cell Signaling #3228) (used at 1:1000), MCM7 (Cell Signaling #4018) (used at 1:1000), DNA polymerase epsilon (Abcam #134941) (used at 1:500), ATR (Cell Signaling #13934) (used at 1:1000), polymerase delta (Santa Cruz #17776) (used at 1:500), RPA34 (Invitrogen #MA1-26418) (used at 1:1000), PCNA (Santa Cruz #56) (used at 1:500), histone H3 (Cell Signaling #9715) (1:1000), H4meK20 (Rockland #600-401-J01) (used 1:500), H3acK9 (Rockland #600-401-I72) (used at 1:500), Chk1 (Cell Signaling #2360) (used at 1:1000), Chk1 pS345 (Cell Signaling #2348) (used at 1:1000), Chk1 p296 (Cell Signaling #2349) (used at 1:1000), GAPDH (Abcam #8245) (used at 1:10,000), 53BP1 (Millipore MAB3802) (used for IF at 1:500), IdU (Becton Dickinson #347580) (used at 1:100 for IF, 1:20 for combing), CldU (Abcam#6326) (used 1:20 for combing). ATRi's were AZD6738 (AstraZeneca), Ve-822 (Selleckchem #S7102), and ETP46464 (synthesis documented previously[50]; Chk1 inhibitors were UCN01 (Sigma), AZD7762 (Selleckchem #S1532); ATM inhibitor Ku55933 (AstraZeneca);

Cdc7 inhibitor PHA-767491 (Selleckchem #S2742); Cdk2 inhibitor CVT-313 (Santa Cruz), EdU (Invitrogen), IdU (Sigma), CldU (MP Biomedical).

**Mass spectrometry.** Nuclease-insoluble chromatin-associated proteins were resolved by denaturing gel electrophoresis. Gels were stained for 10 min with Coomassie and 20 equivalently sized gel slices were excised for each sample and processed by in-gel digestion. Duplicate injections of each gel band digest were analyzed by nanoflow LC (Easy-nLC1000, ThermoFisher Scientific Inc.) coupled online with a hybrid quadrupole Orbitrap MS (Q Exactive, ThermoFisher Scientific Inc.). Following system equilibration, each sample was loaded onto a 2-cm reversed-phase (C-18) vented pre-column (ThermoFisher Scientific Inc.) at 2 µl min$^{-1}$ with mobile phase A (0.1% formic acid in water). Peptides were resolved on a 100 µm I.D. × 360 µm O.D. × 200 mm long capillary column (Polymicro Technologies, Phoenix, AZ, USA) slurry packed in house with 5 µm diameter, 100 Å pore size reversed phase (Magic C18 AQ, Bruker Michrom Bioresources, Auburn, CA, USA) at a constant flow rate of 200 nl min$^{-1}$ by development of a linear gradient of 0.5% mobile phase B (0.1% formic acid in acetonitrile) per min for 80 min, which was then elevated to 95% mobile phase B in 10 min. The column was washed for 10 min with 95% mobile phase B and then equilibrated to 100% mobile phase A prior to the next sample injection. The MS was configured to collect high-resolution ($R = 70,000$ at $m/z$ 200) broadband mass spectra ($m/z$ 375–1800) in profile mode using the lock mass feature for the polydimethylcyclosiloxane ion ($m/z$ 445.12002) generated in the electrospray process from which the ten most abundant peptide molecular ions dynamically determined from the MS scan were selected for tandem MS. Mass spectrometric conditions were set as follows: electrospray voltage, 1.7 kV; no sheath and auxiliary gas flow; capillary temperature, 250 °C; S-Lens RF level, 60%. The automatic gain control (AGC) was set at $1.0 \times 10^6$ with a maximum ion accumulation time of 50 ms. Tandem MS were acquired with the following settings: resolution, 17,500; AGC, $1.0 \times 10^6$; maximum ion accumulation time, 200 ms; isolation window, 3.0 $m/z$; normalized collision energy, 25; underfill ratio, 0.1% ($5.0 \times 10^3$). Dynamic exclusion (40 s) was utilized to minimize redundant selection of peptides for MS/MS.

Tandem mass spectra were searched against the UniProt human protein database (downloaded on 02/21/2014, 68,756 sequences) from the Universal Protein Resource (www.uniprot.org) using Mascot Daemon/Server (v.2.3.2/v.2.3, Matrix Science Inc., Boston, MA) using the automatic decoy search option. The data were searched with a precursor mass tolerance of 10 p.p.m. and a fragment ion tolerance of 0.6 Da. Cysteine carbamidomethylation ($m/z$ 57.021464) was set as a fixed modification and methionine oxidation ($m/z$ 15.994915) was set as a dynamic modification. A maximum of two missed tryptic cleavages were allowed. Identified peptides were filtered using an ion score cutoff of 33 resulting in a false peptide discovery rate of <1% for all peptides identified. PSMs whose sequence mapped to multiple protein isoforms were grouped as per the principle of parsimony. Protein abundance differences were determined by spectral counting.

**DNA combing.** DNA combing was performed using molecular combing system from Genomic Vision. Cells were sequentially incubated with IdU and CldU and embedded in 1% LMP (low melting point) agarose plugs. Plugs were digested with proteinase K for 20 h, melted at 68 °C, and digested with beta-agarase. Silanized coverslips were dipped into the DNA solution and pulled up at 225 µM s$^{-1}$, baked for 2 h at 60 °C, denatured in 1 M NaOH/1.5 M NaCl, washed with PBS, dehydrated with ethanol, and stained with IdU and CldU antibodies.

**iPOND.** iPOND was performed based on refs. [23, 24]. Briefly, cells were cross-linked with 1% formaldehyde solution and incubated in 0.125 M glycine. Cells were permeabilized in 20 mM HEPES pH 7.5, 50 mM NaCl, 3 mM Mg Cl$_2$, 300 mM sucrose, and 0.5% NP40 (IGEPAL), and then the click reaction was performed in 10 mM ascorbate, 2 mM CuSO$_4$, 25 µM biotin azide for 2 h. Cells were suspended in 25 mM NaCl, 2 mM EDTA, 50 mM Tris-HCl pH 8.0, 1% NP40, and protease inhibitors, sonicated briefly and pelleted. Extraction-sonication step was repeated two times. On the third time, the nuclei were extensively sonicated to obtain DNA fragments under 1.5 kb and cleared by centrifugation. The supernatant was incubated with streptavidin agarose beads. Beads were washed five times with 150 mM NaCl, 2 mM EDTA, 50 mM TrisHCl pH 8.0, 0.5% NP40, and protease inhibitors, boiled with Laemmli sample buffer and analyzed by immunoblotting.

**Purification of the nuclease-insoluble chromatin fraction.** Cells were lysed in cytoplasmic lysis buffer 10 mM Tris-HCl (pH = 7.9), 0.34 M sucrose, 3 mM CaCl$_2$, 2 mM magnesium acetate, 0.1 mM EDTA, 0.5% Nonidet P-40, and protease inhibitors for 20 min on ice. Nuclei were collected by centrifugation and washed with cytoplasmic lysis buffer and with cytoplasmic lysis buffer without Nonidet P-40. Nuclei were lysed in 20 mM HEPES (pH = 7.9), 3 mM EDTA, 10% glycerol, 150 mM potassium acetate, 1.5 mM MgCl$_2$, 0.1% Nonidet P-40 and protease inhibitors for 30 min on ice. Chromatin was pelleted by centrifugation and suspended in 150 mM HEPES (pH = 7.9), 1.5 mM MgCl$_2$, 10% glycerol, 150 mM potassium acetate, and protease inhibitors containing universal nuclease for cell lysis (ThermoFisher #88700) and incubated for 10 min at 37 °C on the shaker. Nuclease-insoluble chromatin was pelleted by centrifugation, washed with water, and dissolved in Laemmli sample buffer.

**DNA constructs, siRNA, and transfections**. FLAG-And-1 (WDHD1) from Origene (#RC216745); GINS4-FLAG from Origene (#RC203336) siRNAs against And-1: si-1 AAGCAGGCAUCUGCAGCAUCCdTdT, si-2 AGGAAAA-CAUGCCUGCCACdTdT. Transfections were carried our using Lipofectamine 2000 or Lipofectamine RNAiMax (Invitrogen).

**Purification of the nuclease-insoluble chromatin fraction**. Cells were fractionated essentially as described[51]. Briefly, cells were lysed in cytoplasmic lysis buffer (10 mM Tris-HCl (pH = 7.9), 0.34 M sucrose, 3 mM $CaCl_2$, 2 mM magnesium acetate, 0.1 mM EDTA, 0.5% Nonidet P-40 and protease inhibitors) for 20 min on ice, nuclei were precipitated by centrifugation and washed once with cytoplasmic lysis buffer and once with cytoplasmic lysis buffer without Nonidet P-40. Nuclei were lysed in nuclear lysis buffer (20 mM HEPES (pH = 7.9), 3 mM EDTA, 10% glycerol, 150 mM potassium acetate, 1.5 mM $MgCl_2$, 0.1% Nonidet P-40 and protease inhibitors) for 30 min on ice. Chromatin was pelleted by centrifugation and washed once with nuclear lysis buffer and once with nuclease-incubation buffer (150 mM HEPES (pH = 7.9), 1.5 mM $MgCl_2$, 10% glycerol, 150 mM potassium acetate, and protease inhibitors). Pellets were resuspended in nuclease-incubation buffer containing universal nuclease for cell lysis (ThermoFisher #88700) and were incubated for 10 min at 37 °C on the shaker. Nuclease-insoluble chromatin was pelleted by centrifugation, washed with water and dissolved in Laemmli sample buffer (BioRad #1610737).

**Immunofluorescence and EdU staining**. For immunofluorescent staining, the cells were plated on 8-well glass slides (Falcon). After treatments, cells were washed with PBS, fixed with 4% formaldehyde 10 min at room temperature, permeabilized with 0.5% Triton X-100 in PBS for 20 min, blocked by 3% BSA in PBS with 0.1% Triton X-100 for 1 h, incubated with primary antibodies overnight at 4 °C. Slides were washed with and incubated with secondary antibodies for 1 h in the dark. After washes with 0.1% Triton X-100 slides were mounted with mounting media with DAPI (Sigma) and analyzed.

In case of EdU staining, cells were labeled with 10 mM EdU for 30′. EdU Click—iT Kit (Invitrogen) was used after immunofluorescence staining.

**Immunoprecipitations and immunoblotting**. Cells were lysed in 50 mM Tris-HCl, pH 7.5, 150 mM NaCl, 50 mM NaF, 0.5% Tween-20, 1% NP40, and protease inhibitors for 20 min on ice. Lysates were cleared by centrifugation, and soluble protein was used for immunoprecipitation or mixed with 2× Laemmli buffer and boiled for 10 min. For immunoprecipitation, protein extracts were incubated with M2-agarose beads (Sigma) at 4 °C for 150 min. Beads were washed five times with lysis buffer. Bound proteins were eluted by incubating protein-bead complexes for 120 min at 4 °C in 100 nM FLAG peptide. Proteins were resolved in 4–12% Bis-Tris or 3–8% Tris-acetate gels (Life Technologies), transferred to 0.4 μm nitrocellulose membrane (Bio-Rad) and immunoblotted.

**Data availability**. The data that support the findings of this study are available from the corresponding author upon request.

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

## Acknowledgements

This work was supported by NIH grants RO1 CA204173, P50 CA090440 and P30 CA047904 and the Uniformed Services University of the Health Sciences from the Defense Health Program (HU0001-16-2-0006).

## Author contributions

T.M. and C.J.B. designed the experiments that were completed by T.M. B.H. and T.P.C. performed the MS. S.S. generated essential reagents and M.J.C. provided essential reagents and expertise that was not available in the literature. T.M., T.P.C. and C.J.B. wrote the paper.

## Additional information

**Competing interests:** M.J.C. is a full time employee of AstraZeneca. The remaining authors declare no competing financial interests.

