## [Peer Review File · Nature Communications]

Reviewers' Comments:

Reviewer #1:

Remarks to the Author:

The authors' main claims in this manuscript are:

- ATR inhibition allows unscheduled origin firing and accumulation of DNA replication proteins on chromatin.
- Mcm4 is hyper-phosphorylated following the inhibition of ATR or Chk1 during S phase.
- This phosphorylation is sensitive to the inhibition of Cdc7 or to preliminary exposure to replication stress. And-1 binds GINS and the strength of the binding is increased by inhibition of ATR.
- Putative phosphor-mimicking mutants of GINS4 bind more to And-1. The C-terminal part of And-1 binds GINS.
- The phosphorylations of Chk1 (both the ATR-dependent S345 and the Chk1-dependent S296) are highly dynamic and both quickly disappear following ATR inhibition.
- Cells treated with ATR inhibitors accumulated ssDNA during replication in a Cdc7 dependent manner and are more prone to DSB following exposure to H₂O₂

While in this paper there are several potentially interesting observations, I have reservations on some of results and conclusions. I find necessary that points below are addressed before publication.

Major points to address:

- The authors indicate that Cdc7 plays the main role in Mcm4 phosphorylation and origin firing following ATR inhibition, while CDK plays only a minor role. The paper then focuses only on the role of Cdc7. The inhibition of the kinases is achieved by using inhibitors. An alternative explanation of the results is that the Cdc7 kinase inhibitor is a much better inhibitor than the one for CDK. The authors don't present relevant controls for the efficacy of the CDK inhibitor in the conditions used. The authors reference that Peterman et al., 2010 shows that both CDK and Cdc7 inhibit origin firing when a Chk1 inhibitor is used. In that paper, however, roscovitine is used to inhibit CDK. The authors should try using different inhibitors of CDK to prove that Cdc7 is particularly important in origin firing when ATR is inhibited. In addition, the author should prove that the effect observed using the Cdc7i are not due to inhibition of other kinases (CDKs?) as well. Without this evidence, some conclusions of the paper are not supported.
- The authors show that replication stress blocks origin firing and Mcm4 hyper-phosphorylation. Since the authors show that Cdc7i and depletion of And-1 block Mcm4 hyper-phosphorylation, it's necessary to prove that neither induces replication stress. In particular, And-1 depletion might affect cell biology (in yeast, *ctf4Δ* cells are sick). The authors should show as well that the cell cycle distribution is not affected in And-1 depleted cells. I don't think checking the levels of geminin is a sufficient control. Finally, how the authors explain the difference in geminin levels following And-1 depletion show in Figure 3 and Sup Fig3.
- I find interesting the observation that a ATRi increases the affinity of And-1 to GINS. Can the authors exclude that AND-1 has a higher affinity for GINS once this is incorporated in the Cdc45-MCM-GINS helicase instead? In this case, the increased interaction would depend on the amount of origin firing more than a regulated interaction GINS-And1. To support their thesis, the authors show that a phospho-mimicking mutant of GINS4 binds preferentially to And-1. This observation, however intriguing, means little unless the authors show that GINS is phosphorylated by Cdc7 during replication, and more so following treatment with an ATRi.
- The authors show that fragment 1-814 of And-1 doesn't bind GINS. Can the fragment bind Pol alpha? How can the author exclude incorrect folding of the fragment? In addition would a fragment

missing only the HMG box bind GINS?

- The data in Fig4 are interesting but are not really related to the rest of the story. I think they should be a supplementary figure or incorporated in Fig2

I have also some minor point I would like to be addressed before the publication of the paper.

Minor points:

- Can the authors explain why they see more MCMs on chromatin when they use ATR inhibitors? When cells fire origins, they are simply using the MCMs already loaded on chromatin, so it's puzzling to see an increase. Can the authors exclude that DNA replication forks precipitates more efficiently and are overrepresented on their chromatin pellet?

- I found some inconsistency in the way some the inhibitors are presented, for example Ve-822 and Ve822, UNC-01 and UNC01. In addition, while you mention it several times, did you use the inhibitor VX-970 in any of your figures?

- The authors claim that Mcm4 is phosphorylated by Cdc7 only in yeast. In reality the phosphorylation by Cdc7 appears to require the priming phosphorylation by CDK.

- The authors should reference work done in yeast by the Diffley's and Toczyski's labs elucidating how the S-phase checkpoint regulates the initiation factor Sld3 and the DDK subunit Dbf4 to stop late origin firing in response to replication stress.

- Could the authors indicate a reference for the validation of the use of the anti-IdU in fixed cells as a marker for ssDNA? Could the authors quantify the level of co-localisation between of the signal of anti-EdU and Anti-IdU?

- In Fig3 legend, "E. coli" should be in italics".

Reviewer #2:

Remarks to the Author:

This manuscript reports novel findings with potentially very important implications for cancer treatment. From a mechanistic point of view, ATR-Chk1 phosphorylations are uncovered on the GINS complex, which will surely be of wide interest in the DNA replication and repair fields.

Of translational-clinical significance, ATR kinase inhibitors including the AZD6738 Inhibitor used in these studies, are in clinical trials. Significantly, it is shown here that the presence of ATR inhibitor induces the firing of "unscheduled" replication origins. These can be inhibited by subsequent treatment with a DNA damaging agent. However, if cells are exposed to DNA damage before the inhibitor is applied, the beneficial effect of the combination is lost. This is very important information that should be published as soon as possible.

1. I recommend that the Abstract is rewritten to more clearly bring out the potential clinical implications.

2. I don't like the figure legends so much. They would be better if each started with a title that summarized the message of the data in the figure.

3. Please briefly summarize the types of clinical trials that are underway with ATR inhibitors (types of cancers, damaging agent combinations). That will help put the present findings into practical perspective.

Reviewer #3:

Remarks to the Author:

Review

General Comments

This is an interesting paper describing the effect of ATR kinase inhibition on unscheduled origin firing. In particular, the authors suggest that the sequence of administration of an ATR kinase inhibitor and a DNA damaging agent affects the resultant DNA damage. A strength of this paper is that the study is systematic and investigates inhibitors that are used in the clinic, thus increasing the clinical relevance of the studies. A weakness of the paper is that normal and tumor cells are not compared in either cell culture or in animal model systems. Testing the schedule dependence normal vs. tumor cells or in vivo animal studies would increase the chance that the schedule dependence is of clinical importance. Likewise, it would be worthwhile to determine in some of the key experiments whether there is an affect on cell survival.

Specific Comments

- 1) Pg 4 para 3 and 4: These experiments were performed with an ATR inhibitor, which could have off-target effects. Would the same results be obtained if ATR were transiently knocked down?
- 2) Pg 5 last para: Would the conclusion be stronger if the same results were obtained with Cdc7 knock down?
- 3) Pg 6 para 2: could the authors expand on why treatment with UV and HU but not IR suppressed ATRi-induced hyperphosphorylation of MCM4 in the chromatin?
- 4) Pg 8 last para: It would be interesting to understand the meaning of these DSBs in terms of cell survival. In particular, does Cdc7 inhibition protect cells from the combination of ATRi and hydrogen peroxide?

Response to Reviewers:

Reviewer #1 (Remarks to the Author):

1) "The authors should try using different inhibitors of CDK to prove that Cdc7 is particularly important in origin firing when ATR is inhibited. In addition, the author should prove that the effect observed using the Cdc7i are not due to inhibition of other kinases (CDKs?)"

We believe our data unambiguously show that the ATRi effects on origin firing are DDK-dependent through both Cdc7 inhibition and Dbf4 knockdown (**Figure S5**). The concentrations of the CDK2 and Cdc7 inhibitors that we used completely block EdU incorporation in U2OS cells at 12h after the addition at 10uM concentration (new Supplemental Figure 2D). Thus, both inhibitors are active at these concentrations in these cells, but have a different impact on origin firing. The concentration of the Cdc7 inhibitor that we used is 20 times lower than that required to inhibit CDK2. Other CDKs are not inhibited by the drug. We have, as the reviewer suggested, performed similar experiments with roscovitine and we have added these data to the supplement (**Figure S2E**). Roscovitine does block ATRi-induced MCM4 hyperphosphorylation, but it is unclear which of its substrates (CDK1, CDK2 and CDK5) is responsible for this effect. Certainly, there may be additional kinases participating in this process, but at this time we cannot present a comprehensive analyses of additional kinases. The additional findings that we have generated are important and as such we have added them to the paper. Nevertheless, our conclusion that Cdc7 is particularly important in origin firing when ATR is inhibited was not challenged by the results of these new experiments and is in fact strengthened.

2) The authors should show as well that the cell cycle distribution is not affected in And-1 depleted cells. ... how the authors explain the difference in geminin levels following And-1 depletion show in Figure 3 and Sup Fig3.

We have performed FACS analysis of And-1 depleted cells and added the new data to the supplement (**Figure S4**). We show that the siRNA against And-1 works consistently more efficiently in U2OS cells than 293T cells, which results in stronger G1-phase arrest. This explains the changes in geminin levels.

3) Can the authors exclude that AND-1 has a higher affinity for GINS once this is incorporated in the Cdc45-MCM-GINS helicase instead? ... This observation, however intriguing, means little unless the authors show that GINS is phosphorylated by Cdc7 during replication, and more so following treatment with an ATRi.

New mass spectrometry data confirms that in response to ATR inhibition Sld5 is phosphorylated at its N-terminus in a Cdc7-dependent manner (**Figures 3G,H and Supplemental Figures S3B-G**). We can't exclude that there's more than one mechanism of regulation of the CMG-And-1 interaction, but our data unambiguously shows that Cdc7-dependent phosphorylation of Sld5 is one such mechanism.

4) The authors show that fragment 1-814 of And-1 doesn't bind GINS. Can the fragment bind Pol alpha? How can the author exclude incorrect folding of the fragment? In addition would a fragment missing only the HMG box bind GINS?

We have confirmed that 1-814 of And-1 does bind pol alpha (**Figure S3H**). This confirms that And-1 is folded correctly. The longer C-terminal deletion mimicking yeast Ctf4 was chosen because it had a higher chance of being properly folded based on its resemblance to yeast Ctf4. We have created a construct with the mutant lacking only the HMG box, but this protein failed to be properly expressed. We suspect that it can't be properly folded or is unstable.

5) The data in Fig4 are interesting but are not really related to the rest of the story. I think they should be a supplementary figure or incorporated in Fig2

We have deleted Figure 4 and put one panel into the supplement to clarify any questions of why Chk1 and ATR inhibitors act the same way.

6) Can the authors explain why they see more MCMs on chromatin when they use ATR inhibitors? <...> Can the authors exclude that DNA replication forks precipitates more efficiently and are overrepresented on their chromatin pellet?

We do not think that ATR inhibition causes MCM loading on chromatin. We think changes in topology after MCM activation resulting in MCM encircling one DNA strand lead to MCM moving to benzonase-insoluble fraction of the chromatin. So in a way, yes, replication forks are abundant in this fraction, and there are more replication forks per cell after ATRi treatment, according to our data.

7) I found some inconsistency in the way some the inhibitors are presented, for example Ve-822 and Ve822, UNC-01 and UNC01. In addition, while you mention it several times, did you use the inhibitor VX-970 in any of your figures?

Thank you. VX970=Ve822. Also, we have relabeled everything to make the names consistent.

8) *The authors claim that Mcm4 is phosphorylated by Cdc7 only in yeast. In reality the phosphorylation by Cdc7 appears to require the priming phosphorylation by CDK.*

Thank you for the comment, the text has been edited accordingly.

9) The authors should reference work done in yeast by the Diffley's and Toczyski's labs elucidating how the S-phase checkpoint regulates the initiation factor Sld3 and the DDK subunit Dbf4 to stop late origin firing in response to replication stress.

Thank you for the comment, we have added these references.

10) Could the authors indicate a reference for the validation of the use of the anti-IdU in fixed cells as a marker for ssDNA? Could the authors quantify the level of co-localisation between of the signal of anti-EdU and Anti-IdU?

The reference is added. Our figure shows that there's significant co-localization between EdU and IdU, but quantifying it only makes sense if we expect the level of co-localization to change under certain conditions, which we don't – we just have a change in the number of IdU-stained cells in general, which we quantified.

11) - In Fig3 legend, "E. coli" should be in italics".

Yes, thank you, we have changed it.

Reviewer #2 (Remarks to the Author):

1. I recommend that the Abstract is rewritten to more clearly bring out the potential clinical implications.

We have added clinical significance to the abstract.

2. I don't like the figure legends so much. They would be better if each started with a title that summarized the message of the data in the figure.

We have added titles to the legends.

3. Please briefly summarize the types of clinical trials that are underway with ATR inhibitors (types of cancers, damaging agent combinations). That will help put the present findings into practical perspective.

We have added a brief description of clinical trials in the discussion section.

Reviewer #3 (Remarks to the Author):

1) Pg 4 para 3 and 4: These experiments were performed with an ATR inhibitor, which could have off-target effects. Would the same results be obtained if ATR were transiently knocked down?

In this paper we used 3 different ATR inhibitors and 2 different Chk1 inhibitors, all of them had the same effects on DNA origin firing. The combing data was obtained with only one inhibitor because, during the course of this work, the combing results were additionally published by two other groups who did not go on to investigate the underlying mechanism as we have here. We have tried using siRNAs against ATR, however while in the total cell lysates the ATR levels decrease significantly, in the chromatin fraction they barely decreased at all. We think that whatever little ATR is left after the knockdown acts to prevent unscheduled origin firing, and a complete knockout of ATR is lethal. Therefore we think that using 5 structurally different compounds is enough to exclude the possibility of the non-specific effects.

2) Pg 5 last para: Would the conclusion be stronger if the same results were obtained with Cdc7 knock down?

Human cells do not replicate with a proper Cdc7 knockdown, so this is tricky. We have performed siRNA knockdown of the Cdc7-binding partner Dbf4, that is required for its activity (**Figure S5**), and it confirms our conclusions.

3) Pg 6 para 2: could the authors expand on why treatment with UV and HU but not IR suppressed ATRi-induced hyperphosphorylation of MCM4 in the chromatin?

ATM is the main pathway activated in response to IR, and ATR activity is very low at 30 minutes after IR. We think that high ATR activity after UV or HU is necessary for the observed effect. We have tried looking at ATRi effects 6h

after 5Gy of IR assuming that proper ATR activation after IR takes time, but it still did not block ATRi-induced MCM4

hyperphosphorylation.

We agree that this is a very important observation and we absolutely plan to investigate it further.

4) Pg 8 last para: It would be interesting to understand the meaning of these DSBs in terms of cell survival. In particular, does Cdc7 inhibition protect cells from the combination of ATRi and hydrogen peroxide?

Yes, it is of extreme importance to show that the sequence of treatments affects cell survival. However this paper is more mechanistic and focuses on the mechanistic consequences of ATR inhibition, therefore we chose not to address this question here. The doses of H₂O₂ used for the DSB assay are not really toxic, so a completely different assay needs to be developed to properly study this effect. Using the clinically relevant chemotherapy drugs would require much longer treatment times, and the corresponding long exposures to Cdc7 inhibitor would be toxic as well. This question will definitely be addressed in our further studies after all proper conditions are established.

Reviewers' Comments:

Reviewer #1:

Remarks to the Author:

the paper should now be published.

Reviewer #3:

Remarks to the Author:

The author has addressed several technical comments.

However, a major limitation of this paper that was mentioned in the initial review, was that it was uncertain how these results impact cell survival and whether there is selectivity of this effect for tumor cells compared to normal cells. In the abstract, the authors state, "These data have immediate clinical relevance as ATR inhibitors are in clinical trials in combination with various DNA damaging agents." This clinical relevance cannot be assessed strictly through signaling studies, but should include the effect on clonogenic survival.